# Sex-Specific Behavioral and Molecular Responses to Maternal Lipopolysaccharide-Induced Immune Activation in a Murine Model: Implications for Neurodevelopmental Disorders

**DOI:** 10.3390/ijms25189885

**Published:** 2024-09-13

**Authors:** Jing Xu, Rujuan Zhao, Mingyang Yan, Meng Zhou, Huanhuan Liu, Xueying Wang, Chang Lu, Qiang Li, Yan Mo, Paihao Zhang, Xingda Ju, Xianlu Zeng

**Affiliations:** 1The Key Laboratory of Molecular Epigenetics of Ministry of Education, Institute of Genetics and Cytology, School of Life Science, Northeast Normal University, Changchun 130024, China; xuj391@nenu.edu.cn (J.X.); moy335@nenu.edu.cn (Y.M.); 2Jilin Provincial Key Laboratory of Cognitive Neuroscience and Brain Development, School of Psychology, Northeast Normal University, Changchun 130024, China; zhaorj046@nenu.edu.cn (R.Z.); mingyangyan@nenu.edu.cn (M.Y.); zhoum747@nenu.edu.cn (M.Z.); huanhuanliu211@163.com (H.L.); wangxy045@nenu.edu.cn (X.W.); luc816@nenu.edu.cn (C.L.); liq245@nenu.edu.cn (Q.L.); zhangph807@nenu.edu.cn (P.Z.)

**Keywords:** maternal immune activation (MIA), lipopolysaccharide (LPS), neurodevelopmental disorders, sex differences, synaptic gene expression

## Abstract

Maternal immune activation (MIA) during pregnancy has been increasingly recognized as a critical factor in the development of neurodevelopmental disorders, with potential sex-specific impacts that are not yet fully understood. In this study, we utilized a murine model to explore the behavioral and molecular consequences of MIA induced by lipopolysaccharide (LPS) administration on embryonic day 12.5. Our findings indicate that male offspring exposed to LPS exhibited significant increases in anxiety-like and depression-like behaviors, while female offspring did not show comparable changes. Molecular analyses revealed alterations in pro-inflammatory cytokine levels and synaptic gene expression in male offspring, suggesting that these molecular disruptions may underlie the observed behavioral differences. These results emphasize the importance of considering sex as a biological variable in studies of neurodevelopmental disorders and highlight the need for further molecular investigations to understand the mechanisms driving these sex-specific outcomes. Our study contributes to the growing evidence that prenatal immune challenges play a pivotal role in the etiology of neurodevelopmental disorders and underscores the potential for sex-specific preventative approaches of MIA.

## 1. Introduction

Aberrant brain development has the potential to give rise to a spectrum of neurodevelopmental disorders, which are characterized by various cerebral dysfunctions, cognitive deficits, emotional disturbances, volitional impairments, and behavioral deviations. In addition to the environmental influences experienced by the infant after birth, considerable focus has been directed towards understanding how the maternal environment before delivery impacts the neurodevelopment of the offspring during postnatal growth [1,2]. Research indicates that the pro-inflammatory environment resulting from maternal bacterial infection significantly affects the neurodevelopmental trajectory of the offspring [3].

Maternal immune activation (MIA) has detrimental effects on individual development [4], which is evident not only in the increased incidence of autism spectrum disorders (ASD) in offspring [5], but also in exacerbated behavioral anomalies [6], and delayed neurodevelopmental milestones [7,8]. A meta-analytical review has demonstrated that maternal infection during pregnancy—irrespective of whether it is viral, bacterial, or of another origin—is associated with a 12% increase in the risk of ASD in offspring [9]. MIA can impair the gut barrier function, increasing intestinal permeability and allowing pro-inflammatory substances such as lipopolysaccharide (LPS) to enter the bloodstream, which may exacerbate MIA and potentially affect fetal neurodevelopment. Additionally, MIA has been linked to the onset of schizophrenia-like behaviors [10], cognitive impairments [11], spatial memory deficits [12,13], and various developmental and neurodevelopmental brain disorders [14,15,16].

MIA resulting from bacterial infection has been shown to induce sex-specific disparities in neural development, leading to pronounced behavioral impairments in offspring [17]. These observations are supported by clinical data, which highlight a disproportionately higher incidence of ASD in males compared to females [18]. In experimental research utilizing murine models to investigate anxiety-related behaviors, empirical evidence has revealed that offspring of both sexes, when exposed to lipopolysaccharide (LPS)-induced immune activation, display signs of anxiety. Notably, male offspring exhibit significantly elevated levels of anxiety compared to females, as demonstrated by their performance in the elevated plus maze assay, indicating sex-dependent variations. In contrast, the open field test, an alternative measure for assessing anxiety, does not show significant sex-based differences [17]. Furthermore, exposure to LPS triggers depressive-like behaviors in both male and female offspring, with males uniquely demonstrating enhanced neuronal excitability [19]. However, current research on the molecular basis of sex differences in neurodevelopmental and behavioral abnormalities caused by MIA is limited, with few studies elucidating these phenomena through defined biological markers.

In this study, we investigate the impact of MIA induced by LPS on the neurodevelopmental and behavioral outcomes in a murine model, with a specific focus on sex-specific differences. LPS, a component of the outer membrane of Gram-negative bacteria, is commonly used to model bacterial infection-induced MIA. We examine anxiety-like and depression-like behaviors in the offspring and explore the molecular mechanisms underlying these behaviors, particularly changes in synaptic gene expression and cytokine profiles. By elucidating the sex-specific effects of MIA, our research aims to contribute to the broader understanding of how prenatal immune challenges can lead to neurodevelopmental disorders and to inform the development of sex-specific preventative strategies.

## 2. Results

### 2.1. LPS-Induced MIA Male Offspring Showed Elevated Anxiety in Elevated Plus Maze Test

Pregnant mothers received an intraperitoneal dose of 60 μg/kg LPS or vehicle (Phosphate Buffered Saline, PBS) on embryonic day (E) 12.5. As shown in Figure 1A,B, the administration of LPS led to a statistically significant increase in IL-6 levels in serum at 3 h post-injection (*p* < 0.05), compared with the injection of PBS mothers, confirming effective MIA. Additionally, a notable increase in IL-6 gene expression was observed in the placenta (*p* < 0.01) (Figure 1C). Subsequently, considering that MIA can lead to behavioral abnormalities in offspring, we assessed the progeny from the PBS and LPS groups using the elevated plus maze test, a recognized method for analyzing anxiety-like behaviors. The results indicated that the adult male offspring in the LPS group exhibited an increased number of entries and longer durations in the closed arms compared to offspring in the PBS group, suggesting an escalation in anxiety-like behavior (Figure 1D–F). Conversely, these males exposed to LPS showed a reduction in entries and time spent in the open arms, reinforcing the concept of heightened anxiety (Figure 1D,G,H). Interestingly, no such differences were observed between female offspring in the LPS and PBS groups, indicating that MIA induced by LPS administration may have sex-specific effects on anxiety levels, with more pronounced impacts on male than on female offspring. Taken together, these data demonstrate that LPS-induced MIA promotes anxiety in male offspring.

### 2.2. LPS-Induced MIA Male Offspring Showed Elevated Anxiety in Open-Field Test

To further confirm our initial findings, we extended our investigation with open-field tests on the offspring of the PBS and LPS groups. To further confirm our preliminary findings, we extended the open-field testing to the offspring of both the PBS and LPS groups, again to assess anxiety-like behavior. The findings highlighted that adult male offspring exposed to LPS experienced a notable decrease in the average speed and distance traveled (Figure 2A–C), correlating with a significant reduction in the time they spent in the central zone and an accompanying increase in the time spent in the outside zone (Figure 2A,D–F). Nonetheless, the total distance traveled in the outside zone remained unchanged (Figure 2G), implying an escalation in anxiety-like behaviors among the male offspring. Conversely, no significant behavioral variations were observed between the LPS and PBS groups for adult female offspring, aligning with the outcomes of the elevated plus maze test. In conclusion, MIA triggered by LPS revealed clear sex-specific disparities, with anxiety-like behaviors being markedly more evident in male offspring.

### 2.3. LPS-Induced MIA Male Offspring Showed Elevated Levels of Depression in Tail Suspension Test

Considering the emotional impact of LPS-induced MIA on offspring, we proceeded to evaluate depressive-like behaviors in the progeny from the PBS and LPS groups utilizing the tail suspension test. Our findings revealed that the male offspring in the LPS group had a considerably diminished total mobile time compared to the PBS group males (Figure 3A), indicative of heightened depressive-like behaviors in LPS-exposed males. Moreover, the LPS-exposed males demonstrated a significantly higher maximum movement velocity compared to their PBS counterparts (Figure 3B), suggesting a more vigorous initial struggle. Strikingly, no discernible differences were observed in the total mobile time or maximum movement speed between female offspring in the LPS and PBS groups, indicating that the depressive-like responses to LPS-induced MIA are more pronounced in males. The insights derived from the tail suspension test are instrumental in clarifying how early environmental influences, such as MIA, can differentially mold the neurodevelopmental paths of male and female offspring. This further emphasizes the critical need to integrate sex as a key variable in neuropsychiatric investigative frameworks.

### 2.4. Effects of Maternal LPS Exposure on Maternal Weight and Offspring Development

Following the induction of MIA via LPS, we monitored the subsequent effects on the development of the offspring, with a focus on weight changes as a key indicator of physiological impact. After intraperitoneal injection of LPS in pregnant mice, there was no significant difference in body weight during the first three days. From E15.5 to E18.5, LPS-treated dams showed significantly less weight gain compared to PBS-treated controls, with notable differences at E17.5 (*p* < 0.01) and E18.5 (*p* < 0.001) (Figure 4A). Subsequently, we conducted a count of the number of male and female offspring, and the results showed that compared with the control group, the number of LPS offspring was significantly reduced (*p* < 0.05), which is consistent with the previously observed decrease in the body weight of the mother mice. At the same time, we found a decreasing trend in the number of male offspring and an increasing trend in the number of female offspring in the LPS group, underscoring the impact of maternal LPS exposure on litter size (Figure 4B). Weekly measurements of offspring body weights revealed that LPS-exposed males exhibited increased growth compared to PBS males, especially between weeks 4 and 5 (*p* < 0.0001), while female offspring showed similar growth patterns across groups, with some differences from week 3 (*p* < 0.0001) (Figure 4C). The sex-specific weight progression of offspring uncovers a discernible trend in offspring from the LPS group, indicating an effect on growth kinetics. These findings collectively suggest that maternal LPS exposure, while not causing drastic changes in the sex ratio, may have significant and subtle effects on offspring development (particularly in males) and influence litter size, which is an important consideration in the context of developmental studies.

### 2.5. LPS-Induced MIA Did Not Alter the Size of the Offspring’s Brain

Aberrations in behavior are often associated with modifications in the cerebral cortex’s structure, a significant developmental stage that commences at embryonic day 11 in mice [20]. Administering LPS via intraperitoneal injection at embryonic day 12.5, we triggered physiological alterations with the potential to impact offspring’s cerebral cortical development. Cortical disarray is a defining characteristic of neuropsychiatric disorders [21,22], and MIA is known to exert significant influence on the development of the cortex [23]. This relationship underscores the intricate interplay between early immune challenges and the structural integrity of the brain’s cognitive hubs. Consequently, we scrutinized the brain structures of 8-week-old adult offspring from both the PBS and LPS groups. Our findings indicated no significant discrepancies in the dimensions of brain length, width, or cortical length between the PBS and LPS-exposed offspring, suggesting that LPS-induced MIA did not substantially impact the global morphological development of the brain within the parameters we measured (Figure 5A–D). This divergence in cortical area enlargement among LPS-exposed females, devoid of analogous changes in males or other morphometric assessments, points to a sophisticated and intricate interplay between in utero inflammatory exposures and brain maturation. It signifies that while some morphological facets of the brain may remain unchanged, elements such as cortical area could be significantly altered, with these modifications not readily evident through conventional brain size or length assessments.

### 2.6. LPS-Induced MIA Promoted Significant Alterations in Synaptic-Related Genes of Male Offspring

Given the prior experiments that established increased anxiety-like and depressive behaviors in LPS-exposed male mice without any corresponding changes in cortical morphology, we speculated these effects might stem from shifts in cortical cytokines or alterations in genes tied to neuronal growth. To explore this, we utilized Luminex assays to scrutinize cytokine expression in the cortices of P0 mice from both PBS and LPS groups. Our findings revealed notable decreases in granulocyte-macrophage colony-stimulating factor (GM-CSF) and IL-12p70 levels in males exposed to lipopolysaccharide (LPS), while no such effects were observed in females. Conversely, we noted a decrease in IL-1α, IL-10, TNF-α, IL-1β, IL-12p40, IL-6, and eosinophil chemotactic factor in males, with an opposite trend in females (Figure 6A). Additionally, stark sexual dimorphisms were observed in the expression of macrophage inflammatory proteins MIP-1β, IL-2, and IL-13, alongside increased monocyte chemotactic protein-1 (MCP-1) and IL-5 mRNA in males, and elevated RANTES, keratinocyte-derived chemokine (KC), and granulocyte colony-stimulating factor (G-CSF) in LPS-exposed males, with particularly pronounced increases in G-CSF and KC. These cytokine modulations could affect the expression of genes involved in neuronal growth and synaptic plasticity, and further shape brain development and function, potentially underpinning the observed behavioral traits.

Considering the significant link between cytokine variations and synaptic plasticity [24], we proceeded to explore the effects of LPS on offspring’s brain development in greater detail. Utilizing PCR Array experiments, we scrutinized synaptic plasticity-related genes in the cortices of P0 mice from both the PBS and LPS groups across different sexes. The results indicated significant sex differences in the expression of genes related to synaptic plasticity, such as Ncam1, Grin2b, and Gria2 (Figure 6B). The analysis delineated a marked increase in the LPS-Male group related to the proliferation of glutamatergic synapses and mechanisms underlying long-term potentiation, as opposed to the PBS-Male group. In contrast, when compared with the PBS-Female group, the LPS-Female group showed a significant uptick in PD-L1 expression and activation of the PD-1 checkpoint pathway, which has a significant bearing on cancer biology (Figure 6C,D). These insights highlight the intricate and gender-divergent immunological reactions to MIA. Concurrently, we utilized a Venn diagram to analyze the shared and unique upregulation and downregulation of synaptic genes within the KEGG pathways of the cortex in the offspring of different sexes from the PBS and LPS groups. The findings show a notable upregulation of four synaptic genes in the male LPS offspring’s cortex versus the male P0 PBS group, and a single gene in the female LPS group compared to the female P0 PBS. Conversely, sixteen genes were downregulated in males and one in females, highlighting sex-specific responses to LPS in synaptic gene expression (Figure 6E,F). In brief, early exposure to inflammation can leave a lasting imprint on cognitive function and behavior. Delving into the pathways of cytokine-induced neuronal and synaptic changes may deepen our comprehension of the pathophysiology behind anxiety and depression, opening avenues for the development of sex-specific therapeutic approaches.

## 3. Discussion

The current study extends the field’s understanding of neurobehavioral consequences of MIA induced by LPS in a murine model. We deliberately selected P0 for molecular analysis to capture the immediate responses to MIA, which are pivotal for early neurodevelopment. This approach is justified as it provides an opportunity to identify early biomarkers and sex-specific vulnerabilities that could predict long-term behavioral changes. At P0, the brain is highly active in synaptogenesis, making it a sensitive period to detect initial molecular changes following MIA. These changes may serve as precursors to the emergence of behavioral phenotypes later in life. Our findings not only corroborate but also extend previous research, illustrating the complex relationship between early-life immune challenges and the emergence of neuropsychiatric disorders, with a pronounced emphasis on sex-specific responses [5,25,26]. These results underscore the detrimental impact of MIA on offspring development, which is manifested in the increased incidence of exacerbated behavioral anomalies. The pronounced sex differences in anxiety-like and depressive-like behaviors observed in LPS-exposed male offspring are consistent with clinical observations and highlight the need to consider sex as a pivotal variable in neurodevelopmental research [17,18]. The absence of gross morphological changes in the brain, despite significant behavioral alterations, suggests that MIA effects extend beyond structural abnormalities to encompass functional and molecular changes [14,15,16,27,28,29,30]. This aligns with the multifactorial nature of neurodevelopmental disorders, which are influenced by complex interactions between genetic, environmental, and immunological factors.

The distinct molecular signatures in male and female offspring, including differential cytokine expression and synaptic gene regulation, indicate a sex-specific vulnerability to MIA effects [17,19,31]. These observations are supported by the role of estrogen receptor β and the blood–brain barrier’s protective role against inflammatory mediators, which may contribute to the sex-specific outcomes observed in our study [31,32]. While our study does not directly address therapeutic interventions, the molecular analysis at P0 is strategic to capture early responses that may serve as predictive biomarkers for long-term outcomes. Further research is needed to understand the molecular mechanisms underlying the behavioral changes induced by MIA and how they might inform future therapeutic strategies. The cytokine modulations observed, particularly the upregulation of certain cytokines in LPS-exposed male offspring, indicate a male-specific vulnerability that may be linked to variations in immune system maturation, hormonal balance, and genetic and epigenetic factors [17,19,32,33]. Notably, our study implicates gut barrier dysfunction, potentially exacerbated by elevated IL-6, as a key pathway through which MIA may influence offspring neurodevelopment, highlighting the importance of gut health in pregnancy [34,35]. Furthermore, the distinct neuroinflammatory response observed in female offspring suggests a potentially more resilient or compensatory mechanism against the behavioral changes seen in males. Unraveling the precise mechanisms behind this protective response is a critical area for future research.

Our study offers insights into the sex-dependent neurobehavioral effects of maternal immune activation due to LPS exposure. It adds to the body of knowledge on the role of early inflammatory events in neurodevelopment and calls for a more comprehensive approach to research that includes the consideration of sex differences. This research may inform future studies on the interaction between the immune system and the central nervous system, which may serve as a foundation for future preventative strategies aimed at mitigating the risks associated with MIA. However, it is important to note that while our findings suggest areas for further exploration, they do not directly imply the development of specific therapeutics at this stage.

Our study provides a compelling exposition of the sex-dependent neurobehavioral repercussions of maternal immune activation, stemming from exposure to LPS. The nuanced perspective it offers on the role of early inflammatory events in sculpting neurodevelopmental trajectories is a significant contribution to the field. It compellingly argues for the expansion of current research horizons to encompass the chronicle of these early immune interactions and their reverberating effects on the neuropsychiatric landscape. The study’s findings beckon a reevaluation of existing paradigms, advocating for a sex-conscious methodology in the quest for preemptive and remedial measures against neuropsychiatric disorders. This approach is particularly pertinent given the prenatal immune challenges that may predispose offspring to such conditions. Furthermore, the elucidation of the intricate dialogue between the immune and central nervous systems, through the lens of sex differences, opens avenues for the innovation of bespoke therapeutic interventions. These interventions could be designed to harness the unique biological and immunological nuances of each sex, thereby enhancing the precision and efficacy of treatments.

## 4. Materials and Methods

### 4.1. Animals

The experimental subjects were C57BL/6J mice, managed in strict accordance with the animal care protocols and ethical standards established by the Key Laboratory of Molecular Epigenetics, Ministry of Education, at Northeast Normal University. All mice were housed in standard laboratory cages, with males and females kept separate. The housing environment was controlled with a 12:12 h light–dark cycle, maintained at a constant temperature, and the mice had unrestricted access to food and water.

A total of 30 pregnant C57BL/6J mice, aged approximately 6–8 weeks, were included in the study. Mice were paired for mating in a male/female ratio of 1:2. The detection of a vaginal plug was recorded as day E0.5, marking the commencement of gestation. Pregnant females were housed individually in laboratory cages, and their weight progression was meticulously tracked throughout the gestation period.

The offspring were nurtured by their mothers until weaning on the 21st postnatal day, at which point they were sorted according to sex and litter, ensuring a maximum occupancy of five mice per cage. To control for pseudoreplication, one to two offspring from each litter were included in the study for behavioral and molecular analyses. This approach ensures that the data collected are not influenced by genetic or environmental factors within the same litter.

### 4.2. LPS Injection and Treatment

C57BL/6J mice, aged approximately 6–8 weeks, were paired for mating in a male/female ratio of 1:2. The detection of a vaginal plug was recorded as day E0.5, marking the commencement of gestation. Subsequently, pregnant females were housed individually in laboratory cages, and their weight progression was meticulously tracked throughout the gestation period. On gestational day E12.5, the pregnant mice were weighed, and those in the experimental group were administered a single intraperitoneal dose of Lipopolysaccharide (LPS) at 60 μg/kg, while the control group received an equivalent volume of PBS. After administration, the mice were returned to their designated habitats, and their body weights were systematically recorded daily until parturition. The offspring were nurtured by their mothers until weaning on the 21st postnatal day, at which point they were sorted according to sex and litter, ensuring a maximum occupancy of five mice per cage.

### 4.3. Elevated Plus-Maze Test

The Elevated Plus Maze test, designed to assess anxiety-related behaviors, consists of two open arms (30 cm × 5 cm × 0.25 cm) and two enclosed arms (30 cm × 5 cm × 15 cm), extending from a central platform (5 cm × 5 cm). The maze, constructed from acrylic with white floors and walls, is elevated 60 cm above the ground and features a slight elevation along the edges of the open arms to prevent falls. During the experiment, mice were placed at the center of the maze, facing a closed arm, and their behaviors were observed over a 5-min period. This behavioral assay exploits rodents’ inherent conflict between their exploratory drive and aversion to open, elevated spaces. Key parameters recorded included the number of entries into each arm type (with an entry defined by the placement of both front paws into an arm), the time spent in each arm type, and the total number of arm entries. These data facilitated the calculation of the proportion of time spent and entries into the open arms. After each trial, the maze was cleaned with alcohol to eliminate residual odors, ensuring a neutral testing environment for subsequent subjects. A negative association was found between the time and entries in the open arms and offspring anxiety, where minimal presence indicated higher anxiety. Conversely, more entries and time in closed arms correlated with increased anxiety, showing a preference for safer, enclosed spaces.

### 4.4. Open-Field Test

The open-field test chamber, measuring 50 cm × 50 cm × 40 cm, was strategically situated in a quiet setting. At the start of the test, animals were gently placed in the center of the chamber, and were monitored by both cameras and timing devices. This test is a critical tool for assessing anxiety levels in rodents, evaluating the balance between their curiosity-driven exploratory behavior and the inherent fear of open, novel environments. This dichotomy is reflected in the animals’ natural tendency to explore the central zone, contrasted by their cautious approach to the novel and exposed areas, resulting in predominant use of the peripheral zone and reduced activity in the central zone. Mice exhibiting extensive movement across the chamber, persistent motion, and higher locomotor speeds are considered to show a higher degree of exploratory autonomy and better psychological health. The findings suggest an inverse correlation between the duration spent in the central zone and anxiety levels, as well as between the central zone’s traversed distance and anxiety. Conversely, increased time and distance in the peripheral zone indicate heightened anxiety levels. Each observation period lasted 5 min.

To maintain the integrity of the experimental conditions and prevent cross-contamination of results, the chamber’s inner walls and floor were thoroughly cleaned after each test to remove any residual indicators (e.g., urine, feces, odor) left by the previous subject. This meticulous preparation was essential before introducing the next animal for testing. During the experiment, several parameters were rigorously recorded, including average speed (m/s), total distance traveled (m), time spent in the center zone (s), total distance traveled in the center zone (m), and time in the outside zone (s), providing a comprehensive assessment of each mouse’s behavior and anxiety levels.

### 4.5. Tail Suspension Test

The tail suspension test is a commonly employed behavioral assay in rodents for evaluating depression-like behaviors, predicated on their reactions to an inescapable stressor. The tail suspension test is conducted in a chamber measuring 50 cm × 50 cm × 50 cm, where mice are suspended by their tails using adhesive tape. This setup allows for the observation of behavioral responses over a 6-min period, recorded by a dedicated camera system. The primary behavioral parameter of interest in this test is the duration of immobility, often interpreted as ‘behavioral despair’. This immobility phase is significant as it signifies a cessation of escape-oriented behaviors, akin to a state of depressive-like mood in rodents.

### 4.6. Isolating the Cerebral Cortex of P0 Mice

Euthanize the P0 mice pups humanely with isoflurane anesthesia and decapitation. Carefully remove the skin above the skull, open the skull with fine forceps and scissors, and expose the brain by taking off the skullcap. Under a dissecting microscope (Olympus, BX43, Olympus Corporation, Tokyo, Japan), meticulously separate the cerebral cortex from the brain, avoiding mix-ups with surrounding structures like the hippocampus and thalamus. Collect the separated cortices in cold PBS or an appropriate preservation solution. The isolated cortex can be subjected to further analysis, or the samples can be rapidly frozen in liquid nitrogen and stored at −80 °C for future use.

### 4.7. Measurement of Different Indicators of Brain Structure in Adult Offspring

Following the behavioral assessments, the brains of the adult mice subjected to these experiments were carefully extracted for further analysis. The brains were placed on a neutral white background, typically A4 paper, ensuring clear contrast for imaging. A straightedge was positioned horizontally beneath each brain to provide a scale reference, and photographs were taken for subsequent analysis. Detailed morphometric analyses were performed using ImageJ, a robust image processing tool provided by the NIH. These analyses included measuring the overall length and width of the brains, as well as specific dimensions related to the cerebral cortex, such as its length and surface area. This morphological data, when combined with the behavioral findings, offers comprehensive insights into the potential neurological underpinnings of observed behaviors, particularly those associated with stress, anxiety, and depressive states in the tested mice.

### 4.8. Luminex Multiplex Analysis

In the study involving postnatal day 0 (P0) female and male mice subjected to either PBS or LPS treatments, cerebral cortex homogenates were prepared to assess the inflammatory response through cytokine profiling. The supernatant from each treatment group was carefully extracted for analysis using a multi-cytokine assay. This assay was performed with the Luminex 200 system, employing a specific panel designed for the detection of 23 mouse cytokines (LabEx, LX-MultiDTM-23, Shanghai, China), in strict adherence to the provided manufacturer’s instructions.

The cytokines and chemokines analyzed included GM-CSF, G-CSF, and a range of interleukins (IL-2, IL-3, IL-4, IL-5, IL-6, IL-9, IL-10, IL-13, IL-1β, IL-1α, IL-12 p40, IL-12 p70, IL-17A), tumor necrosis factor alpha (TNF-α), interferon-gamma (IFN-γ), and several chemokines such as eotaxin, monocyte chemotactic protein-1 (MCP-1), macrophage inflammatory proteins MIP-1α and MIP-1β, regulated on activation, normal T cell expressed and secreted (RANTES), and KC.

### 4.9. RNA Extraction and Real-Time PCR

Cerebral cortex from P0 newborn mice was collected for RNA extraction. RNA was isolated using the TRIzol reagent from Invitrogen, adhering to the protocol provided by the manufacturer. Subsequently, complementary DNA (cDNA) was generated from the extracted RNA using a reverse transcription enzyme (Thermo Fisher Scientific, Waltham, MA, USA). This cDNA served as a template for either semi-quantitative or quantitative PCR, employing the Power SYBR Green PCR Master Mix (Thermo Fisher). The amplification and detection of the PCR products were performed on the QuantStudio 3 Real-Time PCR instrument (Applied Biosystems, Foster City, CA, USA). The qPCR data were evaluated using the 2−ΔΔCt method to determine the relative expression levels of the genes of interest. These levels were then standardized against the expression of the housekeeping gene, β-actin, to account for variations in sample loading and integrity.

IL6 forward primer (5′-3′): ACACATGTTCTCTGGGAAATCGT; Reverse primer (5′-3′): AAGTGCATCATCGTTGTTCATACA. β-actin forward primer (5′-3′): AACAGTCCGCCTAGAAGCAC; Reverse primer (5′-3′): CGTTGACATCCGTAAAGACC.

### 4.10. PCR Array

Synaptic plasticity gene expression data were evaluated through the utilization of the Mouse Synaptic Plasticity PCR Array kit (Catalog No. WC-MRNA0139-M, Wcgene Biotech, Shanghai, China), strictly adhering to the instructions provided by the manufacturer. The subsequent analysis was conducted with Wcgene Biotech software, accessible through their online platform at http://www.wcgene.com (accessed on 20 December 2023). Genes exhibiting fold-changes beyond a threshold of 2.0 or below −2.0 were deemed to possess significant biological relevance. A comprehensive list of these genes can be found in the Table 1.

### 4.11. Statistical Analysis

All data were subjected to analysis of variance (ANOVA) using GraphPad Prism 8. Specifically, a two-way ANOVA was employed for data analysis. All results are expressed as the mean ± standard error of the mean (SEM). Subsequent analyses were conducted using Tukey’s post hoc tests for multiple comparisons. A significance level of *p* < 0.05 was considered statistically significant. (*** *p* < 0.001; ** *p* < 0.01; * *p* < 0.05; ns, not significant).

## 5. Conclusions

This study reveals the profound and sex-specific neurobehavioral effects of maternal LPS exposure, highlighting the necessity for sex-stratified research in neurodevelopmental disorders. The distinct cytokine and synaptic gene expression patterns observed in male offspring suggest that early-life inflammation can induce long-lasting changes in brain function and behavior. These findings contribute to the growing understanding of the role of maternal immune activation in shaping offspring neurodevelopment and support the development of sex-specific therapeutic approaches.

## Figures and Tables

**Figure 1 ijms-25-09885-f001:**
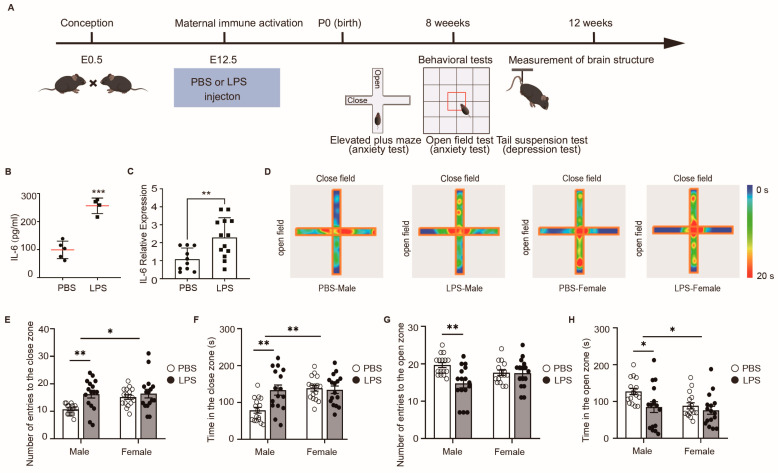
LPS-induced MIA male offspring showed elevated anxiety in elevated plus maze test. (**A**) Schematic of the experimental design. Pregnant C57BL/6J mice were i.p. injected with lipopolysaccharide (LPS) (60 μg/kg) or Phosphate Buffered Saline (PBS) on E12.5 to induce maternal immune activation (MIA). E12.5, Embryonic day 12.5. Red box: the center of the chamber. (**B**) Maternal serum concentrations of IL-6 (PBS, *n* = 5, LPS, *n* = 4) at 3 h after PBS or LPS injection into pregnant dams at E12.5. Unpaired Student’s *t*-tests. *** *p* < 0.001, Mean ± SEM. red line: the mean value. (**C**) Relative IL-6 mRNA expression in E12.5 at 3 h after PBS and LPS injection in placentas of pregnant mice (PBS, *n* = 10, LPS, *n* = 13). Unpaired Student’s *t* test. ** *p* < 0.01. Mean ± SEM. (**D**) Heatmap of mouse movement in the elevated plus maze test in 5 min. (**E**) Number of entries to the close zone. (**F**) Time in the close zone. (**G**) Number of entries to the open zone. (**H**) Time in the open zone. (**E**–**H**) *n* = 16 per group; two-way ANOVA with Tukey’s post hoc tests. * *p* < 0.05, ** *p* < 0.01. Means ± SEM.

**Figure 2 ijms-25-09885-f002:**
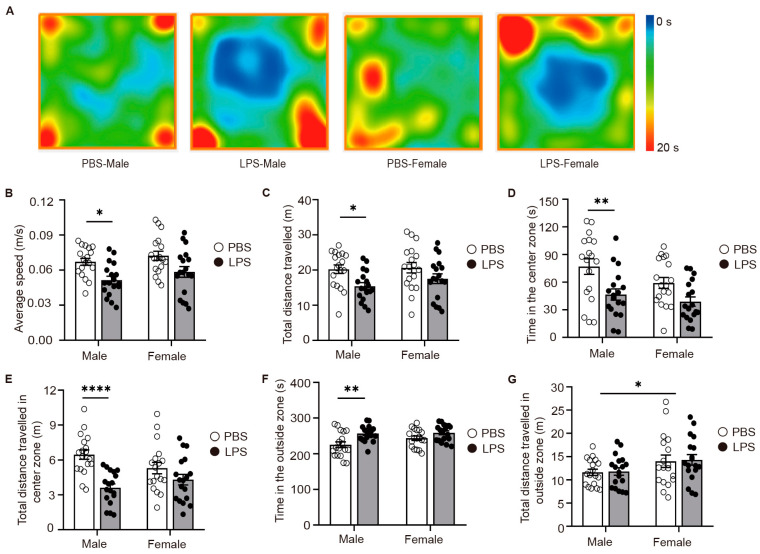
LPS-induced MIA male offspring showed elevated anxiety in open field test. (**A**) Heatmap of mouse movement in the open-field test in 5 min. (**B**) Average speed. (**C**) Total distance traveled. (**D**) Time in the center zone. (**E**) Total distance traveled in the center zone. (**F**) Time in the outside zone. (**G**) Total distance traveled in the outside zone. *n* = 18 per group; two-way ANOVA with Tukey’s post hoc tests. * *p* < 0.05, ** *p* < 0.01, **** *p* < 0.0001. Means ± SEM. LPS, lipopolysaccharide. MIA, maternal immune activation.

**Figure 3 ijms-25-09885-f003:**
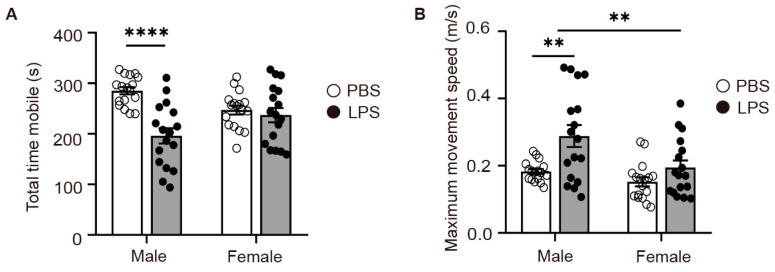
LPS-induced MIA male offspring showed elevated levels of depression in tail suspension test. (**A**) Total time mobile. (**B**) Maximum movement speed. *n* = 17 per group; two-way ANOVA with Tukey’s post hoc tests. ** *p* < 0.01, **** *p* < 0.0001. Means ± SEM. LPS, lipopolysaccharide. MIA, maternal immune activation.

**Figure 4 ijms-25-09885-f004:**
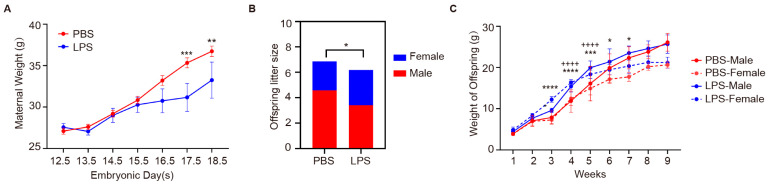
Maternal weight and offspring development in response to LPS exposure. (**A**) Maternal weight changes across embryonic days 12.5 to 18.5 post-PBS or LPS treatment. *n* = 4–19 per group, two-way ANOVA with Tukey’s post hoc tests. ** *p* < 0.01, *** *p* < 0.001. Mean ± SEM. (**B**) Litter size differences between PBS and LPS groups. *n* = 5–7 per group, Mann–Whitney U test. (**C**) Offspring weight progression by sex. *n* = 4–15 per group, two-way ANOVA with Tukey’s post hoc tests. PBS-Female vs. LPS-Female, * *p* < 0.05, *** *p* < 0.001, **** *p* < 0.0001. PBS-Male vs. LPS-Male, ++++ *p* < 0.0001. Means ± SEM. LPS, lipopolysaccharide. MIA, maternal immune activation.

**Figure 5 ijms-25-09885-f005:**
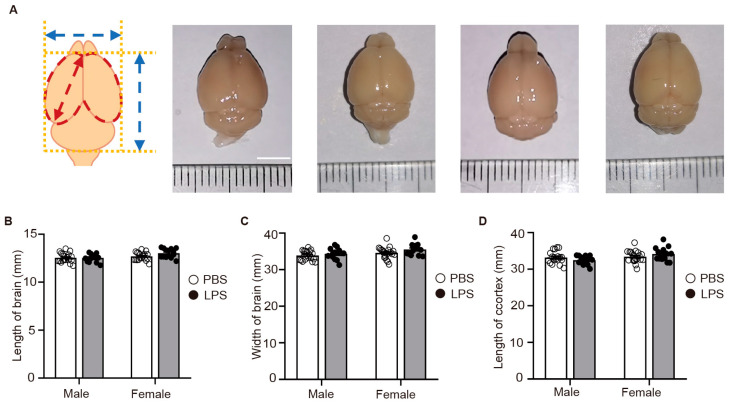
LPS-induced MIA did not lead to noticeable alterations in the size of the offspring’s brain. (**A**) Schematic representation of brain statistics of PBS and LPS offspring. The horizontal blue arrow represents the width of brain, the vertical blue arrow indicates the length of brain, the cortical area is marked with a red line, and the red arrow signifies the length of the cortex. Scale bar = 5 mm. (**B**) Brain of PBS and LPS adult offspring. (**C**) Length of brain. (**D**) Width of brain. E Length of cortex. PBS male, *n* = 20; PBS female, *n* = 20; LPS male, *n* = 14; LPS female, *n* = 14; two-way ANOVA with Tukey’s post hoc tests. Means ± SEM. LPS, lipopolysaccharide. MIA, maternal immune activation.

**Figure 6 ijms-25-09885-f006:**
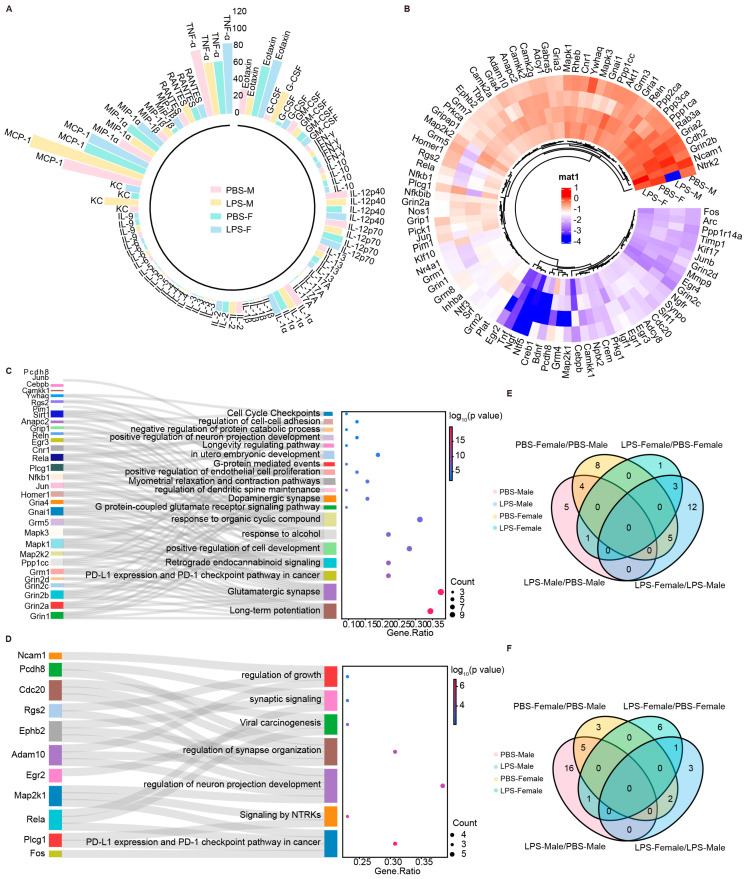
LPS-induced MIA promoted significant alterations in the synaptic-related genes of male offspring. (**A**) Luminex to measure the protein expression levels of various cytokines in the cortex of P0 offspring mice in PBS-M group, LPS-M group, PBS-F group, and LPS-F group (*n* = 3–5 per group). (**B**) PCR array experiments showing differential expression of genes associated with synaptic plasticity in the cortex of P0 offspring of mice of PBS-M, LPS-M, PBS-F, and LPS-F group. (**C**) Selected Gene ontology (GO) annotations enriched in the LPS-M and PBS-M groups. (**D**) Selected Gene ontology (GO) annotations enriched in the LPS-F and PBS-F groups. Bar plot shows the top 10 enrichment score [−log_10_ (Q-value)] of DEGs involving a biological process. (**E**) Venn diagram showing shared and unique upregulated genes in the KEGG pathways within the cortices of PBS and LPS offspring of different sexes. (**F**) Venn diagram showing shared and unique downregulated genes in the KEGG pathways within the cortices of PBS and LPS offspring of different sexes. PBS-M, PBS-Male; LPS-M, LPS-Male; PBS-F, PBS-Female and LPS-F, LPS-Female. LPS, lipopolysaccharide. MIA, maternal immune activation.

**Table 1 ijms-25-09885-t001:** Mouse targets of transcription factor synaptic plasticity-related gene qPCR array.

Mouse	1	2	3	4	5	6	7	8	9	10	11	12
A	Adam10	Adcy1	Adcy8	Akt1	Anapc2	Arc	Bdnf	Camk2a	Camk2g	Camkk1	Camkk2	Cdc20
B	Cdh2	Cebpb	Cnr1	Creb1	Crem	Egr1	Egr2	Egr3	Egr4	Ephb2	Fos	Gabra5
C	Gnai1	Gria1	Gria2	Gria3	Gria4	Grin1	Grin2a	Grin2b	Grin2c	Grin2d	Grip1	Gripap1
D	Grm1	Grm2	Grm3	Grm4	Grm5	Grm7	Grm8	Homer1	Igf1	Inhba	Jun	Junb
E	Kif17	Klf10	Map2k1	Map2k2	Mapk1	Mapk3	Mmp9	Ncam1	Nfkb1	Nfkbib	Ngf	Ngfr
F	Nos1	Nptx2	Nr4a1	Ntf3	Ntf5	Ntrk2	Pcdh8	Pick1	Pim1	Plat	Plcg1	Ppp1ca
G	Ppp1cc	Ppp1r14a	Ppp2ca	Ppp3ca	Prkca	Prkg1	Rab3a	Rela	Reln	Rgs2	Rheb	Sirt1
H	Srf	Synpo	Tbp	Timp1	Tnf	Ywhaq	Actb	Gapdh	Hprt1	B2m	NTC	NTC

## Data Availability

Data are contained within the article.

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
