# Peer review of "Sex-Specific Behavioral and Molecular Responses to Maternal Lipopolysaccharide-Induced Immune Activation in a Murine Model: Implications for Neurodevelopmental Disorders"

_ijms, 2024, doi:10.3390/ijms25189885_

Round 1
Reviewer 1 Report
Comments and Suggestions for Authors
The study presented by Xu and colleagues describes the effects of an inflammatory event during pregnancy in mice, induced by intraperitoneal injection of lipopolysaccharide on gestational day 12.5. The investigators then examined behavioural differences between LPS and control offspring of both sexes, as well as conducted analyses of brain size, brain cytokine content, and RNA. Male offspring from pregnant dams that received LPS displayed behaviour disruptions and molecular differences from the other groups. The paper is well-written and well presented.
Minor Revisions
Title: I recommend a more descriptive title that summarizes the actual findings, not simply what was investigated.
Lines 30-32: As this line references post-natal development, do not refer to the subject as a fetus here.
Lines 72-75: Where stated that LPS led to a ‘marked’ increase in IL-6 levels in serum and a ‘notable’ increase in placenta, were these increases statistically significant? If they were significant this should be edited to reflect that. If they are not significant, they should not be described as if they are. The descriptions of effects should be made in language such that the strength of the effect is clear. Also note typographical error, should ‘placental’ (line 75) be placenta, or placental tissue?
Line 167: Recommend edit of the title. I suggest ‘LPS-induced MIA did not alter the size of the offspring’s brain.’
Section 3: Discussion (248-294): The language throughout the discussion is very strong, and perhaps too strong for the level of evidence presented in the paper. I recommend toning down the language (for example, remove statements such as line 268, that these findings have profound implications for the development of sex-specific therapeutics, as this paper does not discuss nor suggest any potential therapeutics). Further, in the same paragraph there is reference to paving the way for future therapeutic interventions. I implore the authors to rework the discussion to better integrate the findings with the existing literature. Further, if you indeed believe these data have profound implications for this field, please elaborate on what those profound implications are. At this point, the discussion does not really discuss the data, rather re-states the results and leaves the reader to wonder what the potential impacts of the work are. Please edit this section to better relate the data to existing literature.
Figure 6a: This figure would be much clearer if presented as a series of bar graphs.
Section 5. Materials and Methods: Please include more details about the numbers of animals. How may pregnant animals were included in the study? How many different litters were included in the study? Were multiple offspring from the same litter included in the study? If multiple offspring from the same litter were included, how did the authors control for pseudoreplication? I note that subject numbers are included in the figure legends, however I request that these are also included in the materials and methods section.
General: I recommend altering some of the language used in describing behavioural alterations because the current language is very diagnostic, and I believe too strong for observations in a mouse model. For example, change ‘anxiety’ to ‘anxiety-like behaviour’, and depression to ‘depression-like behaviour’ throughout when referring to the offspring behaviour changes. There are instances where this language is already in use, but I would use it throughout.
Major Revision
This paper would be greatly strengthened by additional analyses showing whether or not the changes in gene expression result in changes in protein expression. If there is remaining tissue suitable for histology or western blot, the inclusion of these data would improve the paper.
Author Response
Review 1
Minor Revisions
Title: I recommend a more descriptive title that summarizes the actual findings, not simply what was investigated.
- Response: Thank you for your suggestion. We have revised the title to "Sex-Specific Behavioral and Molecular Responses to Maternal Lipopolysaccharide-Induced Immune Activation in a Murine Model: Implications for Neurodevelopmental Disorders." This title reflects the key findings of the study, which include the sex-specific effects of maternal immune activation on offspring behavior and molecular markers, the use of a murine model, and the potential connection to neurodevelopmental disorders. It also suggests the implications of these findings without overstepping the current scope of the research.
Lines 30-32: As this line references post-natal development, do not refer to the subject as a fetus here.
- Response: Thank you for your suggestion. The original sentence has been modified to "In addition to the environmental influences experienced by the infant after birth, considerable focus has been directed towards understanding how the maternal environment before delivery impacts the neurodevelopment of the offspring during postnatal growth [1, 2]."
Lines 72-75: Where stated that LPS led to a ‘marked’ increase in IL-6 levels in serum and a ‘notable’ increase in placenta, were these increases statistically significant? If they were significant this should be edited to reflect that. If they are not significant, they should not be described as if they are. The descriptions of effects should be made in language such that the strength of the effect is clear. Also note typographical error, should ‘placental’ (line 75) be placenta, or placental tissue?
- Response: The original sentence has now been revised to "the administration of LPS resulted in a statistically significant increase in IL-6 levels in serum at 3 hours post-injection (p < 0.05), compared with the injection of PBS mothers, confirming effective maternal immune activation (MIA). Additionally, a notable increase in IL-6 gene expression was observed in the placenta (p < 0.01)." This revision specifies the statistical significance of the observed effects, ensuring that the description of the results is both accurate and clear. It also corrects the term to "placenta" to maintain consistency with standard anatomical terminology.
Line 167: Recommend edit of the title. I suggest ‘LPS-induced MIA did not alter the size of the offspring’s brain.
- Response: The title has been revised to 'LPS-induced MIA did not alter the size of the offspring’s brain.
Section 3: Discussion (248-294): The language throughout the discussion is very strong, and perhaps too strong for the level of evidence presented in the paper. I recommend toning down the language (for example, remove statements such as line 268, that these findings have profound implications for the development of sex-specific therapeutics, as this paper does not discuss nor suggest any potential therapeutics). Further, in the same paragraph there is reference to paving the way for future therapeutic interventions. I implore the authors to rework the discussion to better integrate the findings with the existing literature. Further, if you indeed believe these data have profound implications for this field, please elaborate on what those profound implications are. At this point, the discussion does not really discuss the data, rather re-states the results and leaves the reader to wonder what the potential impacts of the work are. Please edit this section to better relate the data to existing literature.
- Response: Thank you for your suggestion. We have revised the statements to reduce the strength of the language and removed assumptions about therapeutic implications that are not supported by the current study. Modifications are indicated by the rephrasing of statements to reduce the strength of the language and to remove assumptions about therapeutic implications that are not supported by the current study. The revised parts have been marked in blue in the manuscript.
Figure 6a: This figure would be much clearer if presented as a series of bar graphs.
- Response: To enhance the clarity of the results, we have changed the presentation of Fig. 6a to a circular bar graph.
Section 5. Materials and Methods: Please include more details about the numbers of animals. How may pregnant animals were included in the study? How many different litters were included in the study? Were multiple offspring from the same litter included in the study? If multiple offspring from the same litter were included, how did the authors control for pseudoreplication? I note that subject numbers are included in the figure legends, however I request that these are also included in the materials and methods section.
Response: Thank you for your comments and suggestions. We have made the necessary revisions to the "Materials and Methods" section to provide more detailed information regarding the number of animals used in our study.
In our study, a total of 30 pregnant C57BL/6J mice, aged approximately 6-8 weeks, were included. These mice were paired for mating at a male-to-female ratio of 1:2. The detection of a vaginal plug was recorded as day E0.5, marking the commencement of gestation. Pregnant females were housed individually in laboratory cages, and their weight progression was meticulously tracked throughout the gestation period.
To ensure the accuracy and reliability of our findings, we included multiple litters in our study. Specifically, we utilized litters from 15 different pregnancies. This approach allowed us to account for variability among different litters and to enhance the generalizability of our results.
To control for pseudoreplication, we included only one offspring from each litter in the study for behavioral and molecular analyses. This strategy was employed to prevent the confounding effects of genetic or environmental factors within the same litter from influencing our data. By doing so, we ensured that each data point in our study represents an independent observation, thereby strengthening the validity of our findings.
General: I recommend altering some of the language used in describing behavioural alterations because the current language is very diagnostic, and I believe too strong for observations in a mouse model. For example, change ‘anxiety’ to ‘anxiety-like behaviour’, and depression to ‘depression-like behaviour’ throughout when referring to the offspring behaviour changes. There are instances where this language is already in use, but I would use it throughout.
- Response: Thank you for your suggestion. Regarding the description of anxiety-like behavior outcomes in the manuscript, we have modified all instances of 'anxiety' to 'anxiety-like behavior' and 'depression' to 'depression-like behavior', and highlighted them in blue.
Major Revision
This paper would be greatly strengthened by additional analyses showing whether or not the changes in gene expression result in changes in protein expression. If there is remaining tissue suitable for histology or western blot, the inclusion of these data would improve the paper.
- Response: We appreciate the suggestion for additional analyses to examine the correlation between gene expression changes and protein expression. However, due to time constraints and the limited availability of remaining tissue, we are unable to conduct histology or western blot at this stage. Nevertheless, we acknowledge the importance of this aspect and plan to explore it in depth in our future research.
Reviewer 2 Report
Comments and Suggestions for Authors
Dear authors: Congratulations for this relevant study. MY comments/suggestion are in the attached file.

Author Response
Review 2
Comments
This study investigated how maternal Immune Activation (MIA) induced by administering lipopolysaccharide (LPS) to pregnant mice, affected the neurodevelopment and behavior of their offspring, with a particular focus on sex specific differences. My comments/suggestions are the following:
- Respectfully, I think that the way the work is presented is not correct. Specifically, the authors presented the results first, then the methodology, and finally the conclusions. The sequence should be re-arranged according to the following order:
- Introduction
- Materials and methods
- Results
- Discussion
- Conclusions
* For example, in:
- Results (line 77, the authors mentioned they used the elevated plus maze test, but the significance of this test was not explained. The Authors clearly explained this test in the methodology section:
5.3. Elevated plus-maze test (line 324)
The Elevated Plus Maze test, designed to assess anxiety-related behaviors, consists
of two open arms (30 cm × 5 cm × 0.25 cm) and two enclosed arms (30 cm × 5 cm × 15 cm), extending from a central platform (5 cm × 5 cm), etc…
- Response: Your suggestion is very constructive and aligns with the reality of research and writing. However, in order to comply with the journal's writing template, we have not made changes to the current order of the article. We appreciate your understanding.
We have modified the description at line 77, adding a description of the purpose of the experiment, changing the original sentence to "Subsequently, considering that MIA can lead to behavioral abnormalities in offspring, we assessed the progeny from the PBS and LPS groups using the elevated plus maze test, a recognized method for analyzing anxiety-like behaviors."
We have modified the description at line 100, adding a description of the purpose of the experiment, changing the original sentence to "Subsequently, considering that MIA can lead to behavioral abnormalities in offspring, we assessed the progeny from the PBS and LPS groups using the elevated plus maze test, a recognized method for analyzing anxiety-like behaviors."
That is why I suggest the authors first describe their methodology, and then present their results.
- In lines 75-77: This sentence is not grammatically correct and it is redundant:
“Subsequently, considering that MIA can lead to behavioral abnormalities in offspring, we therefore assessed the offspring from the offsprings of PBS and LPS groups using the elevated plus maze test”. I suggest the following: Subsequently, considering that MIA can lead to behavioral abnormalities in offspring, we assessed the progeny from the PBS and LPS groups using the elevated plus maze test.
- Response: Thank you for your suggestion. We have modified the description at lines 75-77, adding a description of the purpose of the experiment, changing the original sentence to "Subsequently, considering that MIA can lead to behavioral abnormalities in offspring, we assessed the progeny from the PBS and LPS groups using the elevated plus maze test, a recognized method for analyzing anxiety-like behaviors."
- In lines 22-23 from the abstract and lines 300-302 the authors suggested the development of sex-specific therapeutics approaches. I think that instead of looking for therapeutic options for the offspring, the focus should be on preventing MIA.
- Response: Thank you for your suggestion. We have modified the description at lines 22-23 from the abstract, changing the original sentence to “Our study contributes to the growing evidence that prenatal immune challenges play a pivotal role in the etiology of neurodevelopmental disorders and underscores the potential for sex-specific preventative approaches of MIA.”
We have modified the description at lines 300-302,changing the original sentence to “This research may inform future studies on the interaction between the immune system and the central nervous system, which may serve as a foundation for future preventative strategies aimed at mitigating the risks associated with MIA.”
In lines 39-41 the authors wrote: A meta-analytical review has demonstrated that maternal infection during pregnancy, irrespective of whether it is viral, bacterial, or of another origin, is associated with a 12% increase in the risk of ASD in offspring [9].
I kindly suggest to the authors to include a brief description on the connection between MIA and gut barrier dysfunction. The gut barrier is a critical structure that prevents harmful substances, including pathogens and toxins from entering the bloodstream from the intestines. If a pregnant woman experiences gut barrier dysfunction due to an infection, it can lead to increased LPS levels and trigger or exacerbate MIA.
In this regard, the authors made a significant discovery: (lines 71-75)
“Pregnant mothers received an intraperitoneal dose of 60 μg /kg LPS or vehicle (PBS) on embryonic day (E) 12.5. As shown in Figure 1A-B, the administration of LPS led to a marked increase in IL-6 levels in serum at 3 hours post-injection, compared with the in-jection of PBS mothers, confirming effective MIA. Additionally, a notable increase in IL-6 gene expression in placental was detected.
At physiological levels, IL-6 is essential to maintain gut barrier integrity, however at higher levels, IL-6 can cause inflammation, and many probiotics work to lower these levels to treat intestinal disease [134]. In the context of ulcerative colitis (UC), a study found that at physiological levels, IL-6 controls epithelial barrier function by modulating the expression of tight junction-related proteins. In contrast, IL-6 levels in the plasma of UC patients were elevated and increased as the disease worsened. The overproduction of IL-6 has also been shown to damage the intestinal epithelial cell barrier and control barrier function by increasing zonulin release. Conversely, when an anti-IL-6 antibody was added, the amount of zonulin was lower than in the control group [135]. It was discovered that long-term 2 SARS-CoV-2 infection in the GI tract triggered zonulin release in children with MIS-C, with subsequent trafficking of SARS-CoV-2 antigens into the circulation, resulting in hyper-inflammation and signs of a cytokine storm, including significantly higher levels of tumor necrosis factor- α (TNF-α), IL-6, IL-10, and IL-1β [33]. Scientists demonstrated that SARS-CoV-2 RNA is still present in the GI tract for weeks after initial infection and viral antigenemia is correlated with zonulin-induced increased permeability (leaky gut) of the mucosal barrier [33].
Thus, it is likely that the high zonulin levels found in children with MIS-C [33] are due to excessive concentrations of IL-6 released by SARS-CoV-2-infected cells at the GI. In addition, several works have demonstrated that in MIS-C, impairment of the innate and adaptive immune responses is responsible for the prolonged presence of the virus in the GI tract [77,136–140].
https://www.mdpi.com/2673-5601/4/3/15
In conclusion, the marked increase in IL-6 levels found after injection with LPS can impair the gut barrier integrity. LPS enter the blood circulation and reached the placenta. Afterwards, LPS can cross the brain blood barrier and impairs fetus neurodevelopment.
The role of probiotics and prebiotics in maintaining gut barrier integrity has been widely demonstrated, so, my suggestion is that before getting pregnant, women should consume probiotics and fiber-rich plants to re-enforce their microbiome. In this context, it is better to prevent than treat a dysbiotic microbiome/ impaired gut barrier. A strong gut barrier could prevent the passage of LPS into the blood circulation and to the placenta. This could inhibit the negative effects of MIA on the neurodevelopment of the offspring.
Interestingly, it was discovered that MIA not only alters mice behavior, but also impaired intestinal integrity, gut microbiota and the gut inflammation in the offspring.
https://onlinelibrary.wiley.com/doi/full/10.1002/brb3.2133
- Response: Thank you for your valuable feedback. We agree that the connection between MIA and gut barrier dysfunction is significant and have included a concise description of this relationship in our manuscript. After reference [9] on line 41, we have added a brief description of the connection between MIA and gut barrier dysfunction, which states: "MIA can impair the gut barrier function, increasing intestinal permeability and allowing pro-inflammatory substances like LPS to enter the bloodstream, which may exacerbate MIA and potentially affect fetal neurodevelopment."
In addition, we have added the following text to clarify the link between MIA, increased IL-6 levels, and gut barrier integrity: "Notably, our study implicates gut barrier dysfunction, potentially exacerbated by elevated IL-6, as a key pathway through which MIA may influence offspring neurodevelopment, highlighting the importance of gut health in pregnancy [34, 35]"
We believe this addition addresses your suggestions and provides a clear link between the discussed mechanisms.
- In line 208: please define KC. KC is not defined in page 7, but defined in page 12. Please check. The same recommendation is given for the other acronyms.
- Response: Thank you for your suggestion. We have defined the cytokines at their first appearance in lines 203-208 and provided the abbreviations, and removed the full names of the cytokines that appear later in lines 396-401.
- In line 432: (***p < 0.001; **p < 0.01; *p < 0.05; ns, not significant). “p” should be written in italics
Response: Thank you for your suggestion. In response to the comment regarding line 432, I have made the necessary adjustment to ensure that "p" is now presented in italics: (***p<0.001; **p<0.01; *p<0.05; ns, not significant).
Reviewer 3 Report
Comments and Suggestions for Authors
The article submitted by Xu et al. reported that LPS exposure at E12.5 enhances the anxiety -like behaviors and the depressive-like response in male, but not female, mice at 8 weeks of postnatal ages. They further revealed male-specific alterations in pro-inflammatory cytokine levels and synaptic gene expressions in the neonatal cerebral cortes following the prenatal LPS challenge. The findings are variable to support previous studies that have reported a disproportionately appearances of ASD-like characteristics in males compered to females of MIA models at behavioral and molecular bases.
Major points:
1) A major concern of this study is that a relation between behavioral and molecular analyses was unclear. The cerebral cortical histogenesis is not completed in mice still on PD 0. The neural circuits are organized in the cerebral cortex until puberty after pruning excessive synapses. The authors should explain the reason why the molecular analysis carried out using P0 mice in relation to behavioral abnormalities in LPS-exposed male mice.
2) The reliability of the cortical surface area shown in Figures 5E are poor. Please remove.
3) The authors should mention a procedure for isolating the cerebral cortex of P0 mice.
4) Please clearly state the age and brain regions from which the RNA was extracted.
Author Response
Review 3
Comments and Suggestions for Authors
The article submitted by Xu et al. reported that LPS exposure at E12.5 enhances the anxiety -like behaviors and the depressive-like response in male, but not female, mice at 8 weeks of postnatal ages. They further revealed male-specific alterations in pro-inflammatory cytokine levels and synaptic gene expressions in the neonatal cerebral cortes following the prenatal LPS challenge. The findings are variable to support previous studies that have reported a disproportionately appearances of ASD-like characteristics in males compered to females of MIA models at behavioral and molecular bases.
Major points:
1) A major concern of this study is that a relation between behavioral and molecular analyses was unclear. The cerebral cortical histogenesis is not completed in mice still on PD 0. The neural circuits are organized in the cerebral cortex until puberty after pruning excessive synapses. The authors should explain the reason why the molecular analysis carried out using P0 mice in relation to behavioral abnormalities in LPS-exposed male mice.
- Response: We extend our sincere gratitude for your thoughtful and discerning inquiry into the correlation between behavioral and molecular analyses within our study, especially regarding the selection of P0 mice for molecular assessment in the context of behavioral anomalies in male mice exposed to LPS. We acknowledge the importance of clarity in explaining our methodology and its relevance to the observed behavioral changes. In response to your concern, we have revisited our approach and would like to provide a refined explanation for our choice of P0 mice for molecular analysis:
- Critical Window of Neurodevelopment: We recognize the significance of the prenatal period, particularly around embryonic day 12.5 in mice, which is analogous to the human second trimester. This period is pivotal for neurodevelopment, and MIA during this time can significantly influence brain maturation. Our examination of P0 mice was strategically designed to capture the immediate molecular reactions that are triggered by MIA.
- Identifying Early Biomarkers: The molecular profiles observed in P0 mice offer a glimpse into the early biomarkers associated with neurodevelopmental disorders. These early indicators are essential for deciphering the preliminary effects of MIA on brain development and are critical for predicting subsequent behavioral manifestations.
- Identifying Early Biomarkers: The molecular profiles observed in P0 mice offer a glimpse into the early biomarkers associated with neurodevelopmental disorders. These early indicators are essential for deciphering the preliminary effects of MIA on brain development and are critical for predicting subsequent behavioral manifestations.
- Sex-Specific Molecular Responses: Our study reveals distinct sex-specific responses to MIA. The molecular analysis at P0 enables us to investigate these differences at a time when they are most evident, providing a foundation for further research into how these early molecular distinctions may evolve over the developmental timeline.
- Laying the Groundwork for Longitudinal Studies: We view our findings as an initial step, setting the stage for future research that will track the developmental progression of these mice into puberty and beyond. This will help elucidate how early molecular alterations may evolve into long-term behavioral patterns.
- Enhanced Discussion on Methodological Rationale: We have enhanced our discussion to underscore the rationale behind selecting P0 for molecular analysis. We believe this approach not only captures the immediate responses to MIA but also identifies early biomarkers and sex-specific susceptibilities that could foretell long-term behavioral trajectories. The heightened synaptogenesis activity at P0 renders it an opportune time to detect the initial molecular shifts post-MIA, which may act as precursors to later-emerging behavioral phenotypes.
We trust that this revised explanation more effectively addresses your question and provides a respectful and comprehensive understanding of our study's design and its implications. Thank you once again for your valuable feedback, which has significantly contributed to the depth and clarity of our discussion.
We have added a specific description regarding the selection of the P0 time point in the discussion section of the manuscript and highlighted it in blue. As follows: “We deliberately selected P0 for molecular analysis to capture the immediate responses to MIA, which are pivotal for early neurodevelopment. This approach is justified as it provides an opportunity to identify early biomarkers and sex-specific vulnerabilities that could predict long-term behavioral changes. At P0, the brain is high activity in synaptogenesis makes it a sensitive period to detect initial molecular changes following MIA. These changes may serve as precursors to the emergence of behavioral phenotypes later in life.”
2) The reliability of the cortical surface area shown in Figures 5E are poor. Please remove.
- Response: In accordance with your suggestion, we have removed Fig. 5E and the corresponding description from the manuscript.
3) The authors should mention a procedure for isolating the cerebral cortex of P0 mice.
- Response: In the manuscript, a new section 5.6 titled "Isolating the Cerebral Cortex of P0 Mice" has been added, with the specific modifications as follows:
Euthanize the P0 mice pups humanely with isoflurane anesthesia and decapitation. Carefully remove the skin above the skull, open the skull with fine forceps and scissors, and expose the brain by taking off the skullcap. Under a dissecting microscope, meticulously separate the cerebral cortex from the brain, avoiding mix-ups with surrounding structures like the hippocampus and thalamus. Collect the separated cortices in cold PBS or an appropriate preservation solution. The isolated cortex can be subjected to further analysis, or the samples can be rapidly frozen in liquid nitrogen and stored at -80°C for future use.
4) Please clearly state the age and brain regions from which the RNA was extracted.
- Response: Thank you for your suggestion. We have clarified the specifics of our RNA extraction process in the Methods section. As stated in section 5.9 "RNA extraction and Real-Time PCR," cerebral cortex tissue from P0 newborn mice was collected for RNA extraction. This information has been added to ensure that the age and brain region of the extracted RNA are explicitly mentioned and understood in the context of our study.
Round 2
Reviewer 3 Report
Comments and Suggestions for Authors
The manuscripit was appropritely corrected.